# Emotions, Strategies, and Health: Examining the Impact of an Educational Program on Tanzanian Preschool Children

**DOI:** 10.3390/ijerph19105884

**Published:** 2022-05-12

**Authors:** Lauren E. Kauffman, Elizabeth A. Dura, Dina L. G. Borzekowski

**Affiliations:** 1Department of Epidemiology and Biostatistics, School of Public Health, University of Maryland, College Park, MD 20742, USA; kaufla01@umd.edu; 2Department of International Health, Bloomberg School of Public Health, Johns Hopkins University, Baltimore, MD 21205, USA; edura1@jhu.edu; 3Department of Behavioral and Community Health, School of Public Health, University of Maryland, College Park, MD 20742, USA

**Keywords:** media, children, socio-emotional development, food preferences, receptivity, television, video, Tanzania

## Abstract

Around the world, well-produced television programming can engage vulnerable, hard-to-reach audiences by offering informal education and enrichment. *Akili and Me* is an animated children’s educational program available in Sub-Saharan Africa that provides age and culturally appropriate lessons. In 2018, the producers created socio-emotional and health content. This study examines the relationship between children’s exposure to the new *Akili and Me* content and socio-emotional and health outcomes. Participants included low-income school children (mean age 5.32 years, SD = 0.82) from Arusha, Tanzania. Researchers conducted one-on-one baseline and post-intervention surveys with each participant. Over 12 weeks, the children attended afterschool sessions with screenings of *Akili and Me,* with distinct content screened on different days. The research team recorded children’s attendance and assessed children’s receptivity to the program through character identification. Using MLM regression models with data from 411 participants from 10 public schools, the analyses showed that a greater exposure and receptivity to *Akili and Me* predicted improved outcomes scores on the socio-emotional and health outcomes, controlling for sex, age, baseline scores, and general media receptivity (non-*Akili and Me* characters). Contributing to the literature on educational media, this study shows that exposure to an animated program can teach vulnerable preschool children socio-emotional and health content.

## 1. Introduction

Along with physical and cognitive growth, young children develop important socio-emotional and health behavior skills early in life. Acquiring these skills is important. Children need aptitude in recognizing emotions and learning appropriate reactions to different scenarios. Children must be able to describe when they are feeling “mad”, “embarrassed”, “happy”, or “scared” [1]. It has been observed that children as young as 3 years old can recognize sad expressions, and children as young as 6 can identify angry expressions [2]. Very young children can be introduced to strategies to handle difficult situations. If actions are modeled and rewards and punishments are clear, children can anticipate what to do if they feel scared, are being bullied, or need to cooperate to solve a problem [1]. Social cognitive theory and research show that the process of learning is facilitated and considerably shortened with social models [3,4]. In regard to health behaviors, early childhood is the time when children are taught basic hygiene behaviors. Good habits in terms of handwashing, teeth brushing, physical activity, and healthy eating form the basis of a lifetime of healthy living [5]. Basic knowledge influences children’s behaviors and improves health [6].

Despite this evidence on children’s capacity for socio-emotional and health learning and the importance of children’s learning from birth, many parents feel ill-equipped in terms of passing along appropriate knowledge and skills to their young children. This is especially true among low-income families in resource-poor environments [7,8,9].

In low- and middle-income countries, parents may lack the resources to offer preschool children important life skills. Parents’ time and attention are often focused on meeting more basic needs. When categorized into income quintiles, there is a pattern of decreasing proportional preschool attendance as income decreases in low- and middle-income countries [10].

In Tanzania, pre-primary education prepares children for formal primary school and continued education; in these settings, there is a strong emphasis on academic skills such as literacy and numeracy. Often lessons for very young children use traditional rote learning delivered through didactic approaches. Holistic and playful approaches are rare, despite being commonly used and advocated for in other parts of the world [11]. In contrast to other locations, little classroom time is spent in most Tanzanian preschools addressing non-academic topics, such as social, emotional, and physical well-being [12]. Of concern, these topics may not be taught to preschool children at home either; many parents and caregivers feel they lack the expertise and information to pass such lessons on to their children [13]. These factors make our sample ideal candidates for the intervention in this study.

There are strong beliefs and evidence that educational media can play a significant role in preschoolers’ development [4]. Video and mobile applications can facilitate children’s learning of diverse skills and school readiness among young children. Around the world, well-produced television programming can engage vulnerable, hard-to-reach audiences by offering a source of informal education and enrichment [14,15]. While not a replacement for a resource-rich and healthy learning environment, young children can benefit greatly from educational media [16]. Exposure to educational media can result in significant gains among preschool and early school-aged children in a broad range of content areas, including social, emotional, and physical well-being [14,17].

While some global research now evaluates the impact of different media on young children [18], historically, the most researched children’s program is Sesame Street [14,19,20,21]. Longitudinal research conducted in the U.S. shows that children’s early viewing of Sesame Street can lead to encouraging trajectories, lasting well into secondary school and beyond [20,22]. Internationally, preschool children exposed to the various Sesame Street co-productions have made significant gains in their knowledge of letters, numbers, shapes, science, environment, one’s culture, and health- and safety-related practices [14,23,24,25]. In a six-week intervention study conducted with 223 Tanzanian preschool children, significant gains in literacy, numeracy, social development, and emotional development were associated with children’s receptivity of *Kilimani Sesame*, as assessed through the accurate naming of the program’s characters [24].

Ubongo is a relatively new educational media company based in Dar es Salaam, Tanzania [26]. Since 2016, researchers working alongside producers have examined the impact of Ubongo’s pre-primary program *Akili and Me*. A study conducted in Morogoro, Tanzania, used a quasi-experimental design and considered children’s daily exposure to one of five episodes of *Akili and Me* over four weeks [15]. Among 568 children (average age of 4.8 years), researchers found that those in the treatment group did significantly better than those in the control group on post-intervention assessments of drawing, shape knowledge, number recognition, counting, and English skills (but not letter or emotion identification). A second study explored the Kinyarwandan adaptation of *Akili and Me* and occurred outside the capital city of Kigali, Rwanda. In this work, researchers randomized a sample of 402 kindergarten, first, and second graders to see *Akili and Me* or a popular children’s program daily over two weeks. Children with higher levels of *Akili and Me* receptivity did significantly better on eight of the ten outcomes (counting, number identification, shape knowledge, letter identification, color identification, body part recognition, health knowledge, and vocabulary, but not drawing and writing skills) [18].

In 2018, Ubongo’s executives decided to address new content areas. Producers created a second season of *Akili and Me*, substantially changing how the show presented various content. The study presented in this manuscript explores whether this new *Akili and Me* programming more effectively taught preschool children socio-emotional and health lessons. It was hypothesized that exposure to this educational program could improve preschoolers’ socio-emotional and health skills.

## 2. Materials and Methods

The research team prepared culturally and age-appropriate protocols and instruments with a strong consideration of the ethical treatment of vulnerable subjects, children in this case. The study received approval from the Tanzanian Commission for Science and Technology (COSTECH), the Meru District Council, and the University of Maryland’s Institutional Review Board.

In Tanzania, an in-country supervisor hired two separate teams, one for data collection and another for delivering the intervention. All team members came from the Arusha region and had experience conducting research and/or working with young children. The project’s principal investigator led formal training with both teams. This training involved in-person discussions around the ethics of conducting research with vulnerable populations, appropriate consent and assent procedures, and reliable and valid administration of question batteries.

A five-member data collection team conducted interviews with the participating children at baseline and post-intervention. There were spot-checks of these interviews where supervisors would randomly prompt team members to digitally record interviews. The fourteen-member intervention team ensured that the correct video was being shown on the specified day and week. Intervention team members also recorded attendance, noting both study participants and other children in the viewing sessions. As quality control, intervention team members took photographs at the sites every day of the intervention.

### 2.1. Study Participants—Recruitment from Arusha, Tanzania

This study occurred in Arusha, Tanzania. Arusha is one of Tanzania’s 31 regions and is in the northeast of the country). Around 1.7 million people live in this region, in the shadow of Mount Kilimanjaro and Mount Meru.

In Arusha, most adults lack formal education; around a third have completed primary school, and under 1% of adult males and females have completed secondary school [27]. The most common occupations for men involve agriculture (40.7%), skilled manual (24.3%), and unskilled manual work (18.3%); among females, 44% perform unskilled manual labor, 14% skilled manual, and 13% sales and service occupations [27]. Figure 1 offers images of a typical marketplace and a typical school in Arusha.

The poverty rate in Arusha is 48.4%, with 17.1% of the population living in extreme poverty. Only half (49.6%) of the population have improved sanitation facilities with flush toilets or improved latrines, and a majority (60.5%) rely on firewood for cooking instead of electricity. As of 2017, 78% of Tanzanian households owned a mobile phone, and this percentage has been growing [28].

In Tanzania, pre-primary education is available, but not mandatory. Out of all children in the Arusha region in the relevant age range, less than half were enrolled in pre-primary school in 2018 (net enrollment ratio = 47.7%) [29].Countrywide, there are concerns about the quality of pre-primary education. Most teachers lack formal training and often must manage extremely large numbers of students with few resources. According to UNICEF, in Tanzania, the student-to-qualified-teacher ratio at the pre-primary education level is 131:1. This ratio is 24:1 in private schools and 169:1 in public schools [30]. Figure 2 offers images of the pre-primary schools where the research team worked.

For this study, the team recruited 10 public schools based on previous relationships with government and educational administrators in the region. All parents of children aged 5 or 6 years received written and oral information about the study. Parents provided active consent, but data collection was conducted with just 40 children, randomly selected from those who supplied parental consent. However, all children in the community were allowed to come to the screenings.

### 2.2. Survey Administration

Researchers conducted baseline and post-intervention surveys one-on-one with participating children. The surveys were conducted in Kiswahili and took, on average, 24 min. Researchers offered children the option of when and where they would like to do the survey, but practically all surveys happened at the child’s school. Question batteries were interactive, age-appropriate, and often allowed the child to point to responses rather than verbalize answers.

### 2.3. The Intervention

In contrast to studies where intervention materials are delivered as part of the curriculum or during the school day schedule, this research involved an intervention where screenings occurred in an after-school community viewing setting. This situation more closely resembles how viewing might happen outside of a research study where children choose whether to show up. Each participant had autonomy in how much they would see of the programs. It was up to the individual child how often and on what days they would attend viewings.

The intervention involved a weekly viewing schedule where the intervention team showed different content on different days. The intervention played literacy and numeracy content on Mondays and Tuesdays, respectively, and other children’s programming (no *Akili and Me* content) on Wednesdays. Children would see socio-emotional and health content on Thursdays and Fridays, respectively. In a viewing session lasting 40 min, the episodes were played twice. Intervention team members took strict attendance, recording not only the study participants who were present, but also the number of other children present. The intervention lasted 12 weeks. Viewings did not occur on two Wednesdays, Christmas and New Year’s Day.

### 2.4. Measures

The surveys assessed many outcomes; however, this paper focuses on socio-emotional and health skills. The measures were developed using instruments that have been validated with samples of preschool children in low- and middle-income countries, including several sites in sub-Saharan Africa, such as Rwanda [15,18,24,25]. The full survey is available upon request, and sample questionnaire items have been provided in the Appendix A (Table A1).

#### 2.4.1. Socio-Emotional

The socio-emotional measure involved researchers presenting children with six scenarios using an illustration and a simple one- or two-sentence description. These were familiar and culturally recognizable scenarios; examples included a young child falling down and getting hurt, two children sharing a food treat, or a child wanting to play with a friend. For each scenario, the researcher named four emotions and asked the participating child which emotion a featured character in the scenario might be feeling. Then, the researcher asked the child to identify among four new pictures what the best strategy might be to solve the problem in the original scenario. The overall socio-emotional score involved twelve questions, with sub-scores for naming emotions and identifying strategies.

#### 2.4.2. Health

The health measure also had two parts. The first part involved the researcher asking the participating child questions about a range of health behaviors, including washing, teeth-brushing, exercise, and where to go if one feels sick. In the second part, the researcher assessed if the child could distinguish between healthy and not healthy items. The researcher provided two practice examples, using a picture of broiled fish (healthy) and ice candy (not healthy). After ensuring the participating children understood the exercise, the researcher handed the child eight cards with a recognizable/local food or beverage and instructed the child to sort the pictures into two stacks—one healthy and the other not healthy. The health score involved 19 questions (11 health behaviors and 8 foods).

#### 2.4.3. Receptivity and Attendance

To measure exposure to *Akili and Me*, this research used two approaches. The first was receptivity. Receptivity is a reliable and valid way to capture not only what a child has been exposed to, but also what is salient enough for the child to remember [18,31]. As in similar studies of the impact of children’s media, researchers offered the child a picture card with images of characters. The researcher pointed to each image and asked the child to name the character. A point was only given for a fully correct name. *Akili and Me* receptivity involved naming four characters. The research also measured general media receptivity, assessing whether the children could name popular media characters (i.e., Tinga Tinga, Ben Ten, and Dora the Explorer). Including general media receptivity allows the researchers to control for a child’s cognitive ability to name characters.

Intervention team members recorded attendance as an additional way to assess the child’s exposure to *Akili and Me* material, noting how many times a child showed up on each day of the week. This led to an overall number of days attending when *Akili and Me* was airing (Monday, Tuesday, Thursday, and Friday attendance). Additionally, the number and the percentage of each day a child attended (i.e., number of Mondays, percentage of Mondays) were assessed.

### 2.5. Statistical Methods and Data Analyses

The analyses included participants with data from both the baseline and post-intervention surveys, resulting in a total sample of 411 children. The team created scores for each measure. Typically, scores reflected the number of questions answered correctly, apart from some health behavior questions (hand washing and brushing teeth), which were scored on a scale of 0–3.

The analysis was completed using SAS 9.4 software (SAS Institute Inc., Cary, NC, USA). The team calculated the mean and standard deviation for each score. To compare baseline and post-intervention scores, the team used McNemar’s test for the categorical variables (correct versus incorrect) and the Student’s *t*-test for the interval variables. Next, the team estimated multilevel random intercept models predicting the post-intervention scores, using sex, age (in years), baseline scores, general media receptivity, and *Akili and Me* receptivity as fixed effects. The intercept was allowed to vary by school, allowing for partial pooling, as was determined to be appropriate given the intraclass correlation values. The analyses used the *Akili and Me* receptivity measure, but the team also explored similar models using attendance on different days of the week.

## 3. Results

### 3.1. Participants

This sample included 221 (54%) male and 190 (46%) female children (Table 1). Referring to school records, the mean age was 5.32 (SD = 0.82) years with 47 (11%) 3- and 4-year-olds, 200 (49%) 5-year-olds, and 164 (40%) 6-year-old and older children. As reported by the parents, under a fifth of the sample (*n* = 69, 17%) had electricity in their households. Just 86 (21%) indicated that the family had regular access to television (either in their household or at a neighbor or relative’s home). The comparison of the final study sample to the dropout participants indicated no significant differences in gender (*p* = 0.45) or in age (*p* = 0.34).

### 3.2. Intervention

At baseline, fewer than 6% of the participating children could identify any of the *Akili and Me* characters; children could name on average 0.12 (SD = 0.45) characters, with no significant differences by sex or age. After the intervention period, children could name, on average, 3.0 (SD = 1.3) *Akili and Me* characters. There was not a significant difference by sex (t = −0.51, *p* = 0.61) nor age (F (df = 2) = 1.34, *p* = 0.26). At the end of the intervention, 225 (54.7%) could name all four characters, 16.6% three, 11.9% two, and 8.5% one. Just 8.3% of the sample were unable to name any of the characters.

During the intervention’s first week, participating children attended, on average, 2.10 days (SD = 1.94) of the five screening days. Participation on Monday (29%) was lower than on the other days (42% to 47%) during week 1. By the intervention’s 10th week, children attended on average 4.17 days (SD = 1.29) of the five screening days. Attendance during week 10 was generally high for all days (ranging from 80% to 87%). Participating children attended, on average, 40 (SD = 10) out of 58 potential sessions. Viewing increased over the weeks, with 63.3% attending at least one session in Week 1, 97.1% in Week 6, and 97.6% in Week 12. It should be noted that the number of non-study children also increased during the intervention. Figure A1 shows the raw numbers of students attending viewings by days of the week and by intervention week.

In a bivariate analysis of mean baseline and post-intervention scores (with school as a random intercept), there were significant increases in both socio-emotional and health scores and in the strategy and health behavior subcategories (Table 2 and Figure 3).

There was a borderline significant difference for identifying socioemotional strategies, with the lowest mean score among the 5-year-old group and the highest mean score among those in the 6 and older group (F (df = 2) = 2.87, *p* = 0.06). There was also a borderline significant difference for health behaviors (F (df = 2) = 2.59, *p* = 0.08), with older children performing better at baseline. Interestingly, baseline and post-intervention scores for food were not significantly correlated.

Accounting for school nesting and controlling for child’s sex, age, baseline score, and general media receptivity, *Akili and Me* receptivity significantly added to overall socio-emotional and health scores. Table 3, Table 4, Table 5 and Table 6 show the models for the assessed sub-topics and overall scores, highlighting percent gain attributed to *Akili and Me* receptivity and daily attendance. Consistently, baseline scores and *Akili and Me* receptivity appear as the two significant predictors in these models (Table 3 and Table 4). Knowing a child’s sex, age, or their general media receptivity did not predict the various outcomes.

Baseline scores were significant in the strategies, overall socio-emotional score, health behaviors, and overall health score models. Thursday attendance was significant for the emotions model, but not for the strategies or overall socio-emotional score models (Table 5). Friday attendance was significant for the health behaviors and overall health score models, but not for the foods model (Table 6).

## 4. Discussion

This study shows overwhelmingly that children who watched and remembered characters from *Akili and Me* performed better on assessments of socio-emotional and health skills. We estimated rigorous models nested on school, which controlled for the child’s sex, age, baseline score, and general media receptivity. Considering receptivity to *Akili and Me*, children with higher receptivity performed significantly better on the post-intervention assessments for socio-emotional and health measures. When examining the subcategories, *Akili and Me* receptivity predicted a 16.7% gain in naming emotions and a 19.1% gain in identifying strategies. With the health subcategories, *Akili and Me* receptivity predicted an 8.5% gain in health behaviors and an 8.4% gain in food knowledge.

Additionally, children who attended more Thursdays (exposure to *Akili and Me* socio-emotional content) achieved significantly higher scores in naming emotions than children who attended fewer Thursdays. Children who attended more Fridays (exposure to *Akili and Me* health content) had significantly better scores in overall health content than children who attended fewer Fridays.

Great effort was involved in creating this naturalistic and rigorous study. The intervention resembled typical distribution of educational media in community settings. We offered, rather than forced, children to come to group screenings. The study design involved well-trained researchers, simple procedures, solid measures, and stringent analyses. By using receptivity as our measure of exposure, we are confident that we detected what children saw and remembered from watching *Akili and Me*. Our study design of showing distinct content on different days allowed us to see exposure to different content and its impact on various constructs. In fact, more exposure to specific content resulted in specific and significant improvement in an area.

This study occurred in a resource-poor area of Tanzania where children have little access to media. In a location where children had more available media and alternative activities, it is unclear that we would have observed such high attention and receptivity to an educational television program.

This research contributes to the educational media literature in two important ways. First, this work focuses on socio-emotional and health material, rather than the often-studied content areas of literacy and numeracy. We are encouraged that the participating children could learn to identify emotions and strategies for dealing with situations that they may encounter in the future. Related to health, this study made great efforts to include local foods and beverages. We know that learning information and good dietary practices early in life predicts healthier eating and lower obesity in later years [32,33].

A second contribution of this work is it captures data from children in low-income countries. Much of the existing education technology literature considers children from high-income countries using new technology to gain and improve literacy and numeracy skills, although there has been greater inclusion of low- and middle-income countries in recent years [14]. Particularly in resource-poor regions, it is valuable to know if children can learn new skills from traditional and cost-effective broadcast video. While mobile phone and Internet access are growing at substantial rates, around half of Tanzanian children live in poverty. As the government works to improve electricity and Internet access in the school settings [34], we anticipate that, in the next few years, Tanzanian children around the country may be able to access and benefit from shows such as *Akili and Me*, at least in the community setting.

## 5. Conclusions

Media receptivity (character identification) and attendance of *Akili and Me* showings were associated with increases in socio-emotional and health knowledge from baseline to post-intervention. The study findings and increasing attendance by study and non-study children across the intervention weeks can be considered encouraging to the show’s producers and those trying to deliver targeted content to young children.

This research has positive implications for the well-being of young children in resource-poor locations. The production of more culturally specific health and socio-emotional educational cartoons for such regions is supported and recommended based on this research.

## Figures and Tables

**Figure 1 ijerph-19-05884-f001:**
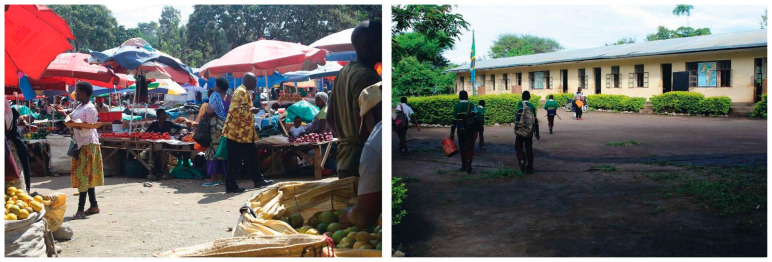
A typical marketplace and a typical school in Arusha, Tanzania.

**Figure 2 ijerph-19-05884-f002:**
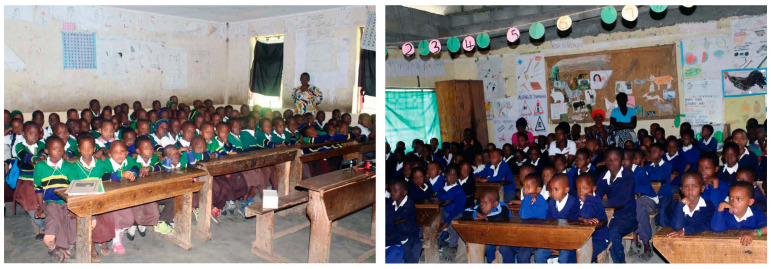
Pre-primary schools in Arusha, Tanzania.

**Figure 3 ijerph-19-05884-f003:**
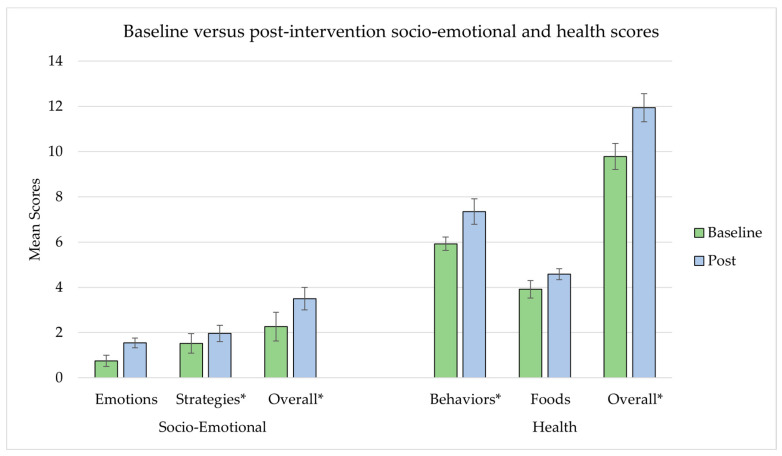
Baseline and post-intervention scores for socio-emotional and health scores. Means and 95% confidence limits for means (bars) calculated accounting for clustering by school. * Indicates significant difference at α = 0.05 in bivariate multi-level model analysis of baseline and post-intervention scores in each category and sub-category.

**Table 1 ijerph-19-05884-t001:** Demographics of participants.

Characteristic	Subcategory	n (%)
Sex	Male	221 (54%)
	Female	190 (46%)
Age	3–4 years old	47 (11%)
	5 years old	200 (49%)
	6–9 years old	164 (40%)
Electricity	Yes	69 (17%)
	No	310 (75%)
	N/A	32 (8%)
Regular access to television	Yes	86 (21%)
	No	230 (56%)
	N/A	95 (23%)

**Table 2 ijerph-19-05884-t002:** Baseline and post-intervention measures.

Variables	Maximum Possible Score	Baseline Score	Post-Intervention Score	Bivariate Multi-Level Model
		Mean (95% CL)	Mean (95% CL)	*p*-Value
**Socio-Emotional**				
Emotions	6	0.74 (0.50, 0.99)	1.54 (1.33, 1.75)	0.2232
Strategies	6	1.52 (1.09, 1.95)	1.96 (1.60, 2.32)	<0.0001
OVERALL	12	2.26 (1.63, 2.89)	3.50 (3.00, 3.99)	<0.0001
**Health**				
Behaviors	19 *	5.92 (5.63, 6.22)	7.35 (6.79, 7.91)	<0.0001
Foods	8	3.91 (3.52, 4.30)	4.58 (4.34, 4.82)	0.5420
OVERALL	27	9.78 (9.21, 10.36)	11.94 (11.32, 12.56)	<0.0001

* There were 11 individual items in the Health Behaviors section. Four of these questions were scored out of three points, resulting in a maximum possible score of 19.

**Table 3 ijerph-19-05884-t003:** Models predicting the socio-emotional outcomes, exploring *Akili and Me* receptivity.

	Emotions	Strategies	Overall Socio- Emotional
	Estimate	Estimate	Estimate
Intercept	0.43	−0.22	−0.2
Sex: Male (vs. Female)	0.36 *	0.01	0.39
Age	0.02	0.11	0.18
Baseline score	0.13 ~	0.49 ***	0.47 ***
General Media Receptivity	−0.53	0.05	−0.52
*Akili and Me* Receptivity	0.25 ***	0.29 ***	0.50 ***
AIC fit statistic	1503.5	1677.9	2002.6
% gain for knowing all four *Akili and Me* characters	16.66%	19.13%	16.74%

~ *p* < 0.10, * *p* < 0.05, *** *p* < 0.001.

**Table 4 ijerph-19-05884-t004:** Models predicting the health outcomes, exploring *Akili and Me* receptivity.

	Health Behaviors	Foods	Overall Health
	Estimate	Estimate	Estimate
Intercept	4.20 ***	3.29 ***	6.48 ***
Sex: Male (vs. Female)	0.14	0.24	0.48 ~
Age	0.04	0.1	0.19
Baseline score	0.29 ***	0.03	0.26 ***
General media receptivity	0.49	−0.55	−0.06
*Akili and Me* receptivity	0.40 ***	0.17 **	0.57 ***
AIC fit statistic	1822.4	1505.9	2053
% gain for knowing all four *Akili and Me* characters	8.45%	8.39%	8.44%

~ *p* < 0.10, ** *p* < 0.01, *** *p* < 0.001.

**Table 5 ijerph-19-05884-t005:** The HLM models predicting the socio-emotional outcomes, exploring Thursday attendance.

	Emotions	Strategies	Overall Socio- Emotional
	Estimate	Estimate	Estimate
Intercept	0.43	0.32	0.34
Sex: Male (vs. Female)	0.38 *	0.04	0.42
Age	0.05	0.14	0.24
Baseline score	0.12 ~	0.53 ***	0.53 ***
General media receptivity	−0.47	0.08	−0.46
Thursday attendance (days)	0.07 *	0.01	0.06
AIC fit statistic	1520.7	1693.6	2024.4

~ *p* < 0.10, * *p* < 0.05, *** *p* < 0.001.

**Table 6 ijerph-19-05884-t006:** The HLM models predicting the health outcomes, exploring Friday attendance.

	Health Behaviors	Foods	Overall Health
	Estimate	Estimate	Estimate
Intercept	4.09 **	3.39 ***	6.51 ***
Sex: Male (vs. Female)	0.09	0.24	0.44
Age	0.05	0.12	0.21
Baseline score	0.33 ***	0.04	0.30 ***
General media receptivity	0.4	−0.54	−0.17
Friday attendance (days)	0.12 *	0.03	0.14 *
AIC fit statistic	1837.5	1514.4	2072.6

* *p* < 0.05, ** *p* < 0.01, *** *p* < 0.001.

## Data Availability

Data can be made available upon request.

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
