# Peer review of "Emotions, Strategies, and Health: Examining the Impact of an Educational Program on Tanzanian Preschool Children"

_ijerph, 2022, doi:10.3390/ijerph19105884_

Round 1
Reviewer 1 Report
Thank you for your paper, I do believe it will make a valuable contribution to the discussion concerning the development of preschool children. Your paper is well written, and interesting. However I do suggest some changes and points to address below:
Firstly, in the introduction part, I think the author should add one paragraph that tell us the situation of social-emotion and health about the preschool children. From this we can see that this research question is very important in the sample areas. If the development of the preschool children in sample areas are very well, we will think that we don't need conduct the intervernion in sampe areas.
Secondly, in the measures part, I think the author should tell readers more imformation about the scales (socio-emotional and health), such as the provenance of the scale, who write the scale? what is the reliability and validity? I think you should introduce briefly the questionnaire items about exact health service needs to reader.
Thirdly, I think the author should tell the readers that how to choose the 10 public preschools. And how to choose the sample children.
Fourthly, the author tell us the gender, age, electricity, regular access to television of the sample. The author should add the household demographics of the sample. For example maternal education, household income and so on. And you should control these variables in table3, 4, 5.
Reviewer 2 Report
Overall, a very well written, interesting study that has high value to children and families in low-income areas but also as a learning tool for those involved in the development of children's media. The introduction provided a good overview of current literature and highlights the context well. I was a bit confused about why Lines 39-42 are included - this doesnt seem to flow from the paragraph before or after. A linking sentence at either end would be beneficial.
The methods are very well described and are appropriate and relevant for this study. The results are well presented and align nicely with the aims of the study. The discussion summarises the findings well and is nicely linked to literature. The concluding comments could be improved - I would like to see this linked to real world application rather than reiterating the findings.
Reviewer 3 Report
The article is about Akili and Me, a lively educational program for children available in Sub-Saharan Africa that offers classes for age and culture. This study examines the relationship between children's exposure to socio-emotional and self health program content and the socio-emotional and health outcomes of Akili and me. It was maked baseline and post-intervention individual survey surveys. Over the weeks, the children participated in after-school sessions with the safety of Akili and me. The study shows that the exposure to an animation program can teach socio-emotional content and health preschool children.
The article is relevant, current and contributions to reflection on the object. It's well written and contributions to world science and the current state of the topic in society. I recommend to ACCEPT the article to publication.
Round 2
Reviewer 1 Report
i don't have any other comments.